# Evaluation of Soil S Pools under 23 Years of Maize Monoculture

Pavel Suran *, Martin Kulhánek [ID], Jiří Balík, Jindřich Černý [ID] and Ondřej Sedlář [ID]

Department of Agro-Environmental Chemistry and Plant Nutrition, Faculty of Agrobiology, Food and Natural Resources, Czech University of Life Sciences, 16500 Prague, Czech Republic; kulhanek@af.czu.cz (M.K.); balik@af.czu.cz (J.B.); cernyj@af.czu.cz (J.Č.); sedlar@af.czu.cz (O.S.)
* Correspondence: suranp@af.czu.cz

**Abstract:** Sulfur nutrition is a critical part of proper crop growth and development. In our study, biomass yields (BY) and S uptake were investigated on long-term maize monoculture on haplic luvisol soil during the 23 years of this trial, as well as changes in water extractable ($S_w$), adsorbed ($S_{ads}$), mineral ($S_{av}$), and pseudo-total S ($S_t$) fractions. Treatments used in this study are: (1) Control (Cont); (2) ammonium sulfate (AS); (3) urea and ammonium nitrate (UAN); (4) UAN + phosphorus and potassium (UAN + PK); (5) UAN + phosphorus, magnesium, sulfur (UAN + PMgS); and (6) Fallow. Recently, the Mehlich 3 method started to be used in the Czech Republic to determine content of plant available S. Using this method, it was found that the content of S extracted by Mehlich 3 ($S_{M3}$) closely correlates to $S_{av}$ in both topsoil and subsoil (r = 0.958 in 1997 and 0.990 in 2019, both at $p < 0.001$). We also found that, on average, during the entire experiment, all treatments had increased yields over Cont (135–147%) and increased S uptake (291, 192, 180, and 246% of Cont for AS, UAN, UAN + PK, and UAN + PMgS, respectively). Examining the changes from 1997 to 2019 in topsoil (0–30 cm depth), we discovered a decrease of S content in $S_w$, $S_{ads}$, $S_{av}$, and $S_t$ fractions on all treatments to an average of 34.6%, 65.8%, 42.2%, and 78.6% of their initial values. The exception was AS treatment, which doubled its initial content in mineral fractions and maintained the same levels of $S_t$, and which we attribute to the very high dose of S on this treatment (142 kg ha$^{-1}$ year$^{-1}$). Using the simple balance method, AS and UAN + PMgS treatments lost 142.2 and 95.3 kg S ha$^{-1}$ year$^{-1}$ to other sinks, except plant uptake, from the entire soil profile (0–60 cm) during 23 years of experiment. Other treatments also show significant losses with the exception of Fallow. Given these results, it is clear that content of sulfur in soil is generally decreasing and attention should be paid mainly towards minimizing of its losses.

**Keywords:** corn; mineral fertilizing; soil sulfur fraction; sulfur balance; Mehlich 3

## 1. Introduction

Sulfur is a key nutrient in plant nutrition, where it appears in the synthesis of amino acids, glutathione, chlorophyl, and other processes [1], influencing yields, crop quality, harvest quality, and other parameters [2,3].

Sulfur deficiencies in plant nutrition in European countries were described by Lehmann et al. [4] and Yang et al. [5], who mention that, in recent decades, the content of S in soils is being reduced due to desulfurization technologies in industry, the use of high analysis fertilizers with low amounts of S [6], and N only fertilization, which increases yields and creates greater demands for plant available nutrients [7]. More recent research states, that amount of wet + dry S depositions are roughly around 18 kg ha$^{-1}$ year$^{-1}$ for Europe [8] and 15 to 30 kg S ha$^{-1}$ year$^{-1}$ in the Czech Republic [9], but currently, in 2019, the total depositions of S in the Czech Republic average 4.2 kg ha$^{-1}$ [10], while in the 1980s, depositions reached 100 kg S ha$^{-1}$ year$^{-1}$ [11], which could have been a sufficient supply for crop production or, even in some cases, toxic for some crops. Because of the above-mentioned facts, S fertilizing is often underestimated [4,12].

The content of total sulfur is divided between organic and inorganic S fractions. The most important for plant nutrition is inorganic $SO_4^{2-}$. Inorganic $SO_4^{2-}$ comprises about 5–10% of the total content [12,13]. The mineral form of S is present in soil in three groups– readily available sulfates in soil solution, which represents usually only about 1% of total S; adsorbed sulfates on soil particles that can be released into soil solution and directly influence the preservation of the sulfate balance with the soil solution [14,15]; and sulfates occluded in calcium and magnesium carbonate precipitates, unavailable to plants. The majority of total S is organic bound. This fraction can play a role in resupplying mineral sulfates throughout the vegetation [16,17], which is a biologically mediated process [18].

The total sulfur in soils at 10 experimental stations in the Czech Republic averaged 221 mg kg$^{-1}$ in 1981, and slightly decreased to 204 mg kg$^{-1}$ in 2007. A much bigger decrease (by 50%) was discovered in plant available S content, mainly because of lower S inputs from atmospheric deposition [9]. Kulhánek et al. [19] investigated the influence of long-term fertilization in stationary field experiments under crop rotation (from 1996 to 2014), and they describe a decrease of mineral S fraction in soil by up to almost 40% on unfertilized treatment on two different stations. On the other hand, it was found that the lowest decrease was observed on farmyard manure and sewage sludge treatments.

Inorganic sulfates are very mobile in soil solution and are susceptible to leaching [20]. Due to higher atmospheric emissions in the past, sulfur cycles in soil were altered by high S inputs and leaching of sulfates from soils was observed [4]. With lowering depositions from the atmosphere, a recovery of S cycles was noted, but it lags behind the decrease of S depositions [21]. On top of that, there is evidence that a significant part of the sulfates present in streams is microbiologically cycled [4] and this mineralization of organic S is responsible for the delayed reduction of sulfates in streams [22].

Sulfates can be supplied either with organic fertilizers, (e.g., farmyard manure, slurry, and others), or mineral fertilizers, e.g., commonly used S fertilizers ammonium sulfate (AS) or magnesium sulfate. AS is water soluble [6] and quite mobile in soil [23–25]. Dijksterhuis and Oenema [23] noticed that a leachate in their pot experiment fertilized with AS had a higher concentration of $SO_4^{2-}$ anion with higher losses on sandy soil in comparison to clayey soil. Riley et al. [24] set up an experiment with soil of natural structure in a monolithic lysimeter fertilized by AS with a dose of 50 kg S ha$^{-1}$. They found that 72% of S was lost due to leaching. Bergholm and Majdi [25] described a similar problem in field trials, where 684 kg S ha$^{-1}$ supplied as the sum of atmospheric depositions and S fertilizer only showed a loss of 339 kg S ha$^{-1}$ by leaching. Kieserite ($MgSO_4$) is also water soluble and shows similar tendencies to AS. Ponette et al. [26] noticed significant vertical transport of $SO_4^{2-}$ anions in soil after application of kieserite. In the trials of Frank and Stuanes [27], 21% of S added in 35 kg ha$^{-1}$ of kieserite-S was leached below 80 cm soil depth. Generally, these authors mention that drier conditions result in less leaching than wet conditions.

Maize (*Zea mays* L.) shows a good yield response to sulfur fertilization for doses around 25 kg S ha$^{-1}$ [28] and up to 60 kg S ha$^{-1}$ [29]. Requirements for S fertilization can be influenced by soil organic matter content ($C_{SOM}$) [28], from which S can be released to supply plants [16,17]; the availability of other nutrients that can increase biomass production and, therefore, S uptake [30,31], water availability [32], and genetic gains [7].

Changes of water extractable, adsorbed, and plant available S were previously investigated by Kulhánek et al. [19]. Gourav et al. [33] conducted a similar experiment with the inclusion of organic and total sulfur. Both of these experiments were conducted under crop rotation.

The aim and contribution of this study is to investigate the changes of S fractions and S leaching under somewhat "extreme" conditions of 27 years of continuous maize monoculture. This study aims to assess the effects of mineral fertilization on long-term stationary trials, focusing on: sulfur content in soil, balance of total sulfur inputs and outputs in soil, and influence of fertilizers on biomass yields and S uptake.

## 2. Materials and Methods

### 2.1. Setup and Background

A long-term stationary experiment with maize monoculture was established in 1993 in Červený Újezd in the Czech Republic, as part of the Czech University of Life Sciences' experimental fields. Characteristics of the experimental site can be found in Table 1. The field experiment was conducted in a randomized complete block design with an area of 170 m$^2$ plot size. All variants were replicated four times. The maize plants were planted on each plot, except fallow. The maize hybrids DK 205 (1993–1996), Torena (1997 and 1998), DK 254 (1999), Compact (2000), Etendard (2001–2003), Rivaldo (2004–2011), RGT Indexx (2012–2014), and RGT Sixxtus (2015–2019) were planted on each plot at a density of 80,000 plants per ha.

**Table 1.** Characteristics of the Červený Újezd experimental site.

| | |
|---|---|
| GPS coordinates | 50°4′22″ N<br>14°10′19″ E |
| Altitude (meters above sea level) | 410 |
| Mean annual precipitation (mm) | 493 |
| Mean annual temperature (°C) | 7.7 |
| Soil type | Haplic luvisol |
| Soil texture | Loam |
| pH (CaCl$_2$) | 6.5 |
| Clay (%) (<0.002 mm) | 5.4 |
| Silt (%) (0.002–0.05 mm) | 68.1 |
| Sand (%) (0.05–2 mm) | 26.5 |
| Bulk density topsoil (g cm$^{-3}$) | 1.47 |
| Bulk density subsoil (g cm$^{-3}$) | 1.50 |
| C$_{SOM}$ (%) | 1.26 |
| Cation exchange capacity (mmol$_{(+)}$/kg) | 118 |

### 2.2. Treatments

For the purpose of this study, several treatments were selected, namely: (1) unfertilized control with crop production (Cont); (2) ammonium sulfate (AS); (3) urea and ammonium nitrate (UAN); (4) UAN + phosphorus and potassium (UAN + PK); (5) UAN + phosphorus, magnesium, sulfur (UAN + PMgS); and (6) unfertilized fallow without crop production (Fallow). Annual nutrient inputs from fertilizers are described in Table 2. N fertilizers were applied in spring before sowing. P, K, Mg, and S fertilizers were applied in autumn. Each of the fertilizers was applied in a single dose. No additional amendments were added to individual treatments, and only stubble from previous year was incorporated to the soil. No analysis of S content in stubble was performed. The entire harvest of above-ground biomass was removed from the trial fields and all treatments were later ploughed (including Fallow). Total sulfur inputs including annual sulfur deposition (dry + wet) are described in Table 3. Precipitation (as an only source of irrigation) was measured directly at the experimental site; however, there is no equipment for detecting S depositions. Therefore, we used data provided from the meteorological station Prague Ruzyně, which is the nearest professional station measuring S depositions (the distance is about 10 km by air; GPS: 50°6′0.6″ N, 14°15′19.8″ E).

**Table 2.** Experimental design and annual nutrient dose.

| Treatment | kg ha$^{-1}$ Year$^{-1}$ | | | | |
|---|---|---|---|---|---|
| | **N** | **P** | **K** | **Mg** | **S** |
| Cont | 0 | 0 | 0 | 0 | 0 |
| AS | 120 | 0 | 0 | 0 | 142 |
| UAN | 120 | 0 | 0 | 0 | 0 |
| UAN + PK | 120 | 30 | 150 | 0 | 0 |
| UAN + PMgS | 120 | 30 | 0 | 60 | 84 |
| Fallow | 0 | 0 | 0 | 0 | 0 |

AS—ammonium sulfate (21% N; 24% S), UAN—urea ammonium nitrate solution (30% N), PK—triple superphosphate (21% P) + potassium chloride (50% K), MgS—kieserite ($MgSO_4$; 15% Mg; 21% S).

**Table 3.** Total sulfur inputs (deposition + fertilizer) during experiment.

| Input Period | Cont | AS | UAN | UAN + PK | UAN + PMgS | Fallow |
|---|---|---|---|---|---|---|
| | | | kg S ha$^{-1}$ | | | |
| 1993–1996 | 80 | 648 | 80 | 80 | 416 | 80 |
| 1997–2001 | 61 | 771 | 61 | 61 | 481 | 61 |
| 2002–2007 | 48 | 900 | 48 | 48 | 552 | 48 |
| 2008–2013 | 37 | 889 | 37 | 37 | 541 | 37 |
| 2014–2019 | 29 | 881 | 29 | 29 | 533 | 29 |
| Sum 1993–2019 | 254 | 4088 | 254 | 254 | 2522 | 254 |
| Sum 1997–2019 | 175 | 3441 | 175 | 175 | 2107 | 175 |

Numbers at Cont, UAN, UAN + PK, and fallow represent the net wet + dry S deposition. Data were provided by meteorological station Prague Ruzyně belonging to the Czech Hydrometeorological Institute.

*2.3. Plant Analyses*

Two rows of maize aboveground biomass (20 m$^2$ per plot) were harvested at silage maturity (roughly 65% biomass moisture content, BBCH 75) to obtain the dry aboveground biomass yield (BY). Representative subsamples were harvested and chopped using mechanical chopper weight and subsequently dried in a forced-air oven to constant weight at 40 °C for at least 72 h and were then fine milled (Retsch SM100, Haan, Germany). Briefly, the aliquote 0.25 g of milled sample was weighed and immersed in nitric acid (7 mL of 65% $HNO_3$) and hydrogen peroxide (2 mL of 30% $H_2O_2$). Samples were then digested in a microwave-assisted high-pressure environment. The whole procedure is further described in Tlustoš et al. [34].

*2.4. Soil Analyses*

Archived soil samples of topsoil (0–30 cm depth) and subsoil (30–60 cm depth) sampled after the crop harvest were air-dried and sieved (<2 mm). For available sulfur fractions, a sequential extraction method by Morche [35] and modified by Kulhánek et al. [13] was selected. Briefly, samples were extracted with demineralized water (1/10 *w/v*) to extract the readily available S ($S_w$) fraction and, subsequently, with 0.032 mol/L $NaH_2PO_4$ to extract the adsorbed sulfur ($S_{ads}$) fraction. The sum of these fractions is then the bioavailable sulfur ($S_{av}$). Usually, extraction by 1 mol/L HCl follows extraction of $S_{ads}$, but this fraction is usually measured using ion chromatography (IC), since optical emission spectroscopy with inductively coupled plasma (ICP-OES) tends to be less accurate as it measures a significant part of organic S in the HCl extract, especially on non-calcareous soils, like the soil used in this study. Therefore, HCl extraction was omitted.

The pseudo-total sulfur ($S_t$) concentration in the soil was determined by modified ISO: 11,466 1995 [36] method using *Aqua regia*. The modification was a microwave assisted high pressure digestion and evaporation of samples using a heating plate (150 °C) and, subsequently, the quantitative transfer with distilled water was allocated to a final volume of a 25 mL glass tube, topped up by deionized water, and kept at laboratory temperature

until measurements were taken. Mehlich 3 extraction was also performed following Mehlich [37] in order to also evaluate the $S_{M3}$ fraction.

Sulfur concentrations in all digests and extracts were determined using the optical emission spectroscopy with inductively coupled plasma (ICP-OES) and with axial plasma configuration, Varian, VistaPro, equipped with autosampler SPS-5 (Mulgrave, Australia). The operating measurement wavelength for ICP-OES was 180.7 nm for S.

Statistical analyses were performed using one-way or two-way analysis of variance (ANOVA; Tukey HSD post-hoc test $p < 0.05$) using the SAS® system, (Cary, CA, USA). Even though trials started in 1993, we evaluated the period of 1997–2019 since a more representative dataset is available.

## 3. Results

### 3.1. Biomass Yield and S Uptake

Statistical analysis of average yield in the period 1993–2019 (Figure 1a) shows significant differences between the unfertilized Cont treatment and fertilized treatments, which do not show any differences between each other.

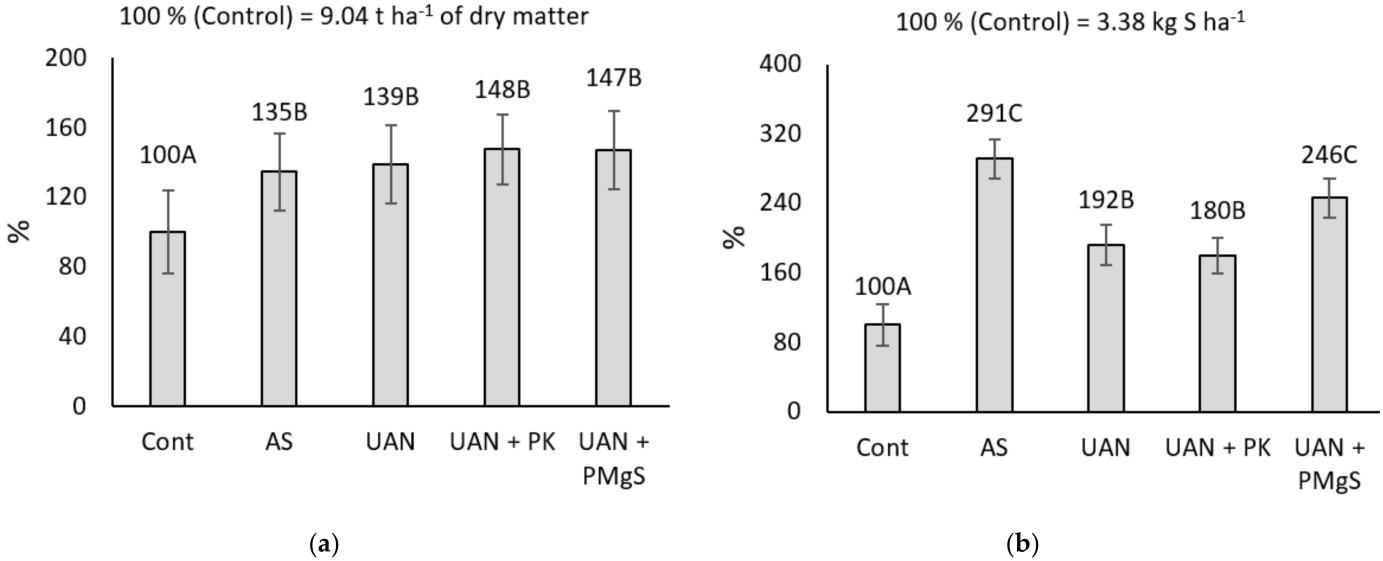

(**a**)        (**b**)

**Figure 1.** Relative average yields (**a**) and sulfur uptake (**b**) at harvest in the years 1993–2019; the capital letters describe statistically significant differences between treatments. Tukey HSD test ($p < 0.05$).

Average sulfur uptake in the same period showed more differences. Fertilized treatments produced significantly higher uptake over the Cont. This time, however, significant differences were present even among the fertilized treatments (as described in Figure 1b), where AS and UAN + PMgS fertilized treatments were comparable to each other, but were higher than UAN and UAN + PK.

A more detailed comparison of S uptake is provided in Table 4. In the period 1993–1996 (judging by the capital letters), differences were already present between the Cont treatment and AS and UAN + PMgS treatments, while UAN and UAN + PK were comparable with Cont. In later periods, UAN and UAN + PK always showed significant increase over Cont. The AS treatment achieved the highest values of S uptake and was always different from other three treatments mentioned. UAN + PMgS was different from UAN and UAN + PK treatments in 1997–2001 and UAN + PK in the 2002–2007 period. AS and UAN + PMgS were not different in any period.

**Table 4.** Detailed plant uptake of sulfur during the experiment (kg S ha$^{-1}$).

| Uptake Period | Cont | AS | UAN | UAN + PK | UAN + PMgS |
|---|---|---|---|---|---|
| | Mean ± SD | Mean ± SD | Mean ± SD | Mean ± SD | Mean ± SD |
| 1993–1996 | 15.7 ± 0.54 A | 39.6 ± 1.39 C | 25.3 ± 1.10 AB | 22.6 ± 0.97 AB | 31.6 ± 0.56 BC |
| 1997–2001 | 24.8 ± 0.67 A | 63.5 ± 1.23 C | 42.7 ± 0.65 B | 40.1 ± 0.70 B | 55.4 ± 0.83 C |
| 2002–2007 | 22.2 ± 0.49 A b | 65.9 ± 1.66 D c | 43.2 ± 1.38 BC b | 40.1 ± 1.14 B b | 56.7 ± 1.90 CD b |
| 2008–2013 | 17.6 ± 0.56 A ab | 58.9 ± 3.04 C b | 39.8 ± 1.74 BC b | 36.8 ± 1.05 B ab | 50.5 ± 1.89 BC ab |
| 2014–2019 | 16.1 ± 0.55 A a | 47.8 ± 1.57 C a | 32.3 ± 1.23 B a | 30.3 ± 1.20 B a | 37.6 ± 1.07 BC a |
| Sum 1993–2019 | 96.3 | 275.6 | 183.3 | 169.9 | 231.7 |
| Sum 1997–2019 | 80.6 | 236.1 | 158 | 147.3 | 200.1 |

The capital letters describe statistically significant differences among treatments. The lowercase letters describe statistically significant differences among years. Tukey HSD test ($p < 0.05$). Only periods of 2002–2007, 2008–2013, and 2014–2019 were compared as these periods last the same number of years.

In order to estimate what fraction was the most important for plant nutrition, a Spearman´s correlation analysis was performed comparing the S biomass content of maize with individual S fractions in topsoil (Table 5). It was possible to compare only the latest values from the year 2019 ($n = 20$). In case of this study, S fertilizers were applied for several years before the first evaluations were performed, causing a shift from a purely linear relationship to a monotonic.

**Table 5.** Spearman's correlation coefficients comparing the relationship between individual S fractions in topsoil and plant S content.

| | $S_W$ | $S_{ads}$ | $S_{M3}$ | $S_{av}$ | $S_t$ | df |
|---|---|---|---|---|---|---|
| Plant S content | 0.674 ** | 0.747 ** | 0.579 ** | 0.725 ** | 0.288 | 18 |

Values marked with ** are significant at $p < 0.01$. Degrees of freedom (df) were the same for each fraction ($n = 20$).

### 3.2. Topsoil Sulfur Content

In topsoil, a two-way ANOVA was performed to analyze the effect of six treatments and three time periods on S content in individual fractions. Results of this analysis (Table 6) showed that both treatment and year had significant effects on content of S in individual fractions. Significant interaction of (year × treatment) was also present for all fractions except $S_t$. To evaluate pairwise differences, a Tukey HSD post-hoc test at $p < 0.05$ was used. This test revealed that interaction of (AS treatment × year 2019) within any of the $S_w$, $S_{ads}$, $S_{M3}$, and $S_{av}$ fractions was significantly different from any other (treatment × year pair) of the same fraction, with the exception being $S_t$, where the interaction was insignificant. Interactions of other pairs were omitted because they do not appear significant nearly as often. Additionally, given the number of possible pairs, this report would become unbearable in volume at 6 treatments and 3 years per each of 5 fractions.

**Table 6.** Results for two-way ANOVA comparing the effects of treatment, year, and their interaction on S content in individual S fractions in topsoil.

| S Fraction | Treatment | | Year | | Treatment × Year | |
|---|---|---|---|---|---|---|
| | F-Value | df | F-Value | df | F-Value | df |
| $S_W$ | 28.07 * | 5 | 19.68 * | 2 | 16.63 * | 10 |
| $S_{ads}$ | 27.08 * | 5 | 3.88 * | 2 | 11.27 * | 10 |
| $S_{M3}$ | 36.17 * | 5 | 4.72 * | 2 | 16.27 * | 10 |
| $S_{av}$ | 30.35 * | 5 | 17.72 * | 2 | 16.94 * | 10 |
| $S_t$ | 3.926 * | 5 | 23.77 * | 2 | 2.08 | 10 |

Values marked with asterisk (*) describe a significant effect at $p < 0.05$.

Influence of individual effects was described using one-way ANOVA. In samples, $S_w$, $S_{ads}$, $S_{av}$, and $S_t$ content were measured (Figures 2 and 3) in the years 1997, 2008, and 2019,

as well as $S_{M3}$ (Figure 4). It is evident that the content of all S fractions was significantly higher in 1997 than in 2019. There are two exceptions, however. Firstly, the content of $S_t$ in 2019 and 1997 on the Fallow treatment (Figure 3b) is not significantly different. Secondly, the content of $S_t$ in 1997 and 2019 on the AS treatment is not significantly different.

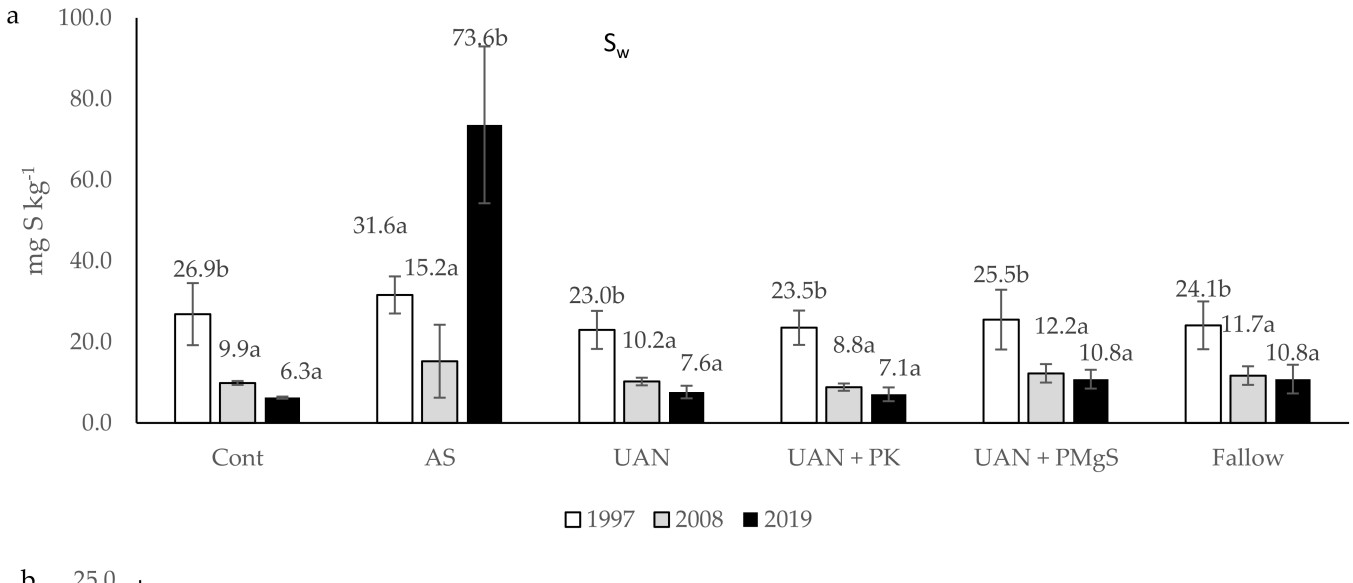

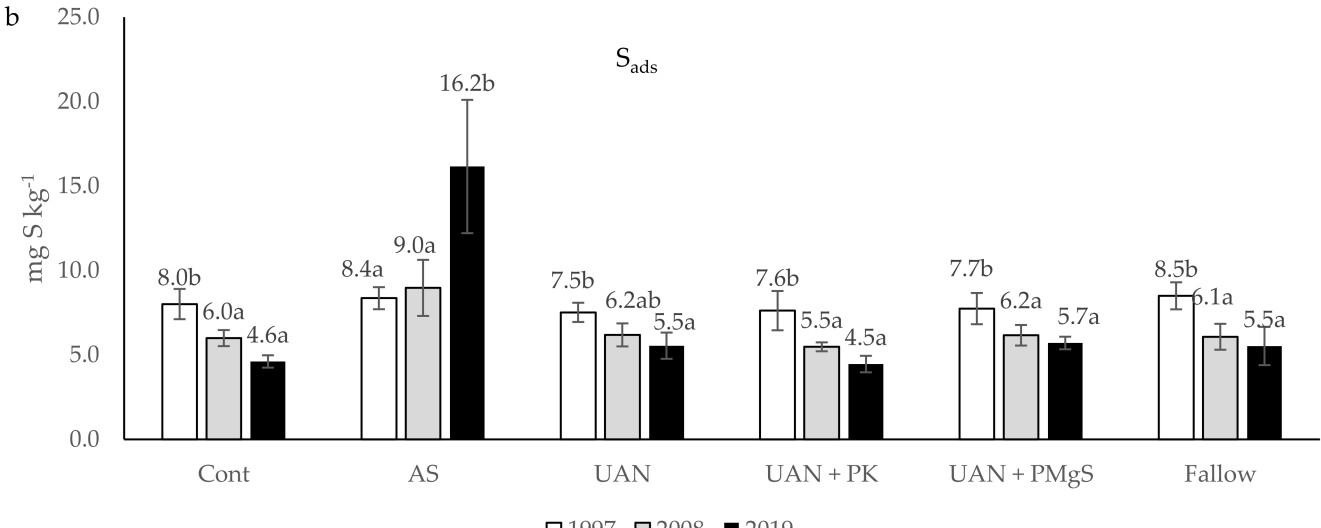

**Figure 2.** Content of sulfur fractions in topsoil (mg S kg$^{-1}$): $S_w$ (**a**); $S_{ads}$ (**b**). Comparison of the effect of year within treatment; the lowercase letters describe statistically significant differences between years. Tukey HSD test ($p < 0.05$).

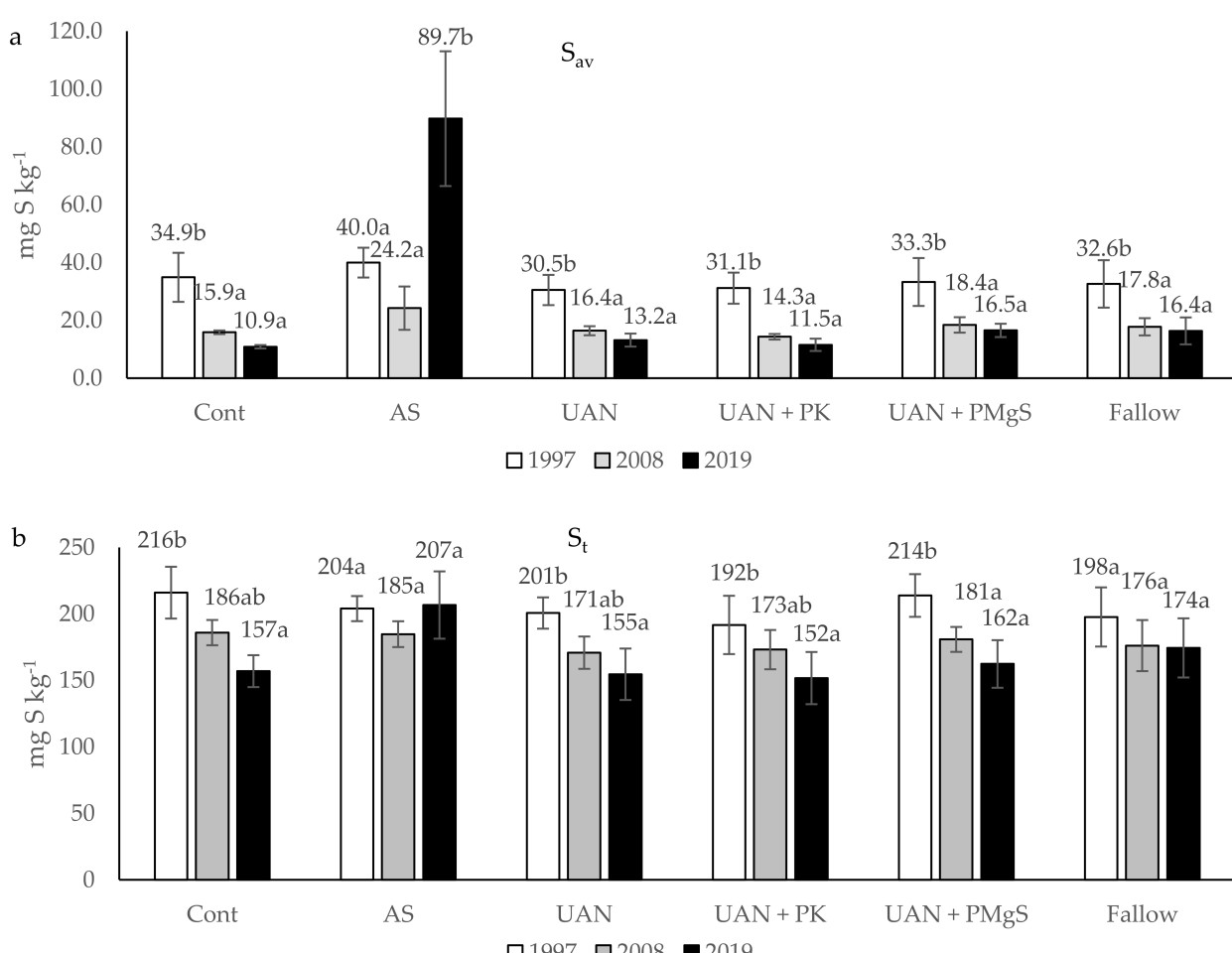

**Figure 3.** Content of sulfur fractions in topsoil (mg S kg$^{-1}$): S$_{av}$ (**a**); S$_t$ (**b**). Comparison of the effect of year within treatment. The lowercase letters describe statistically significant differences between years. Tukey HSD test ($p < 0.05$).

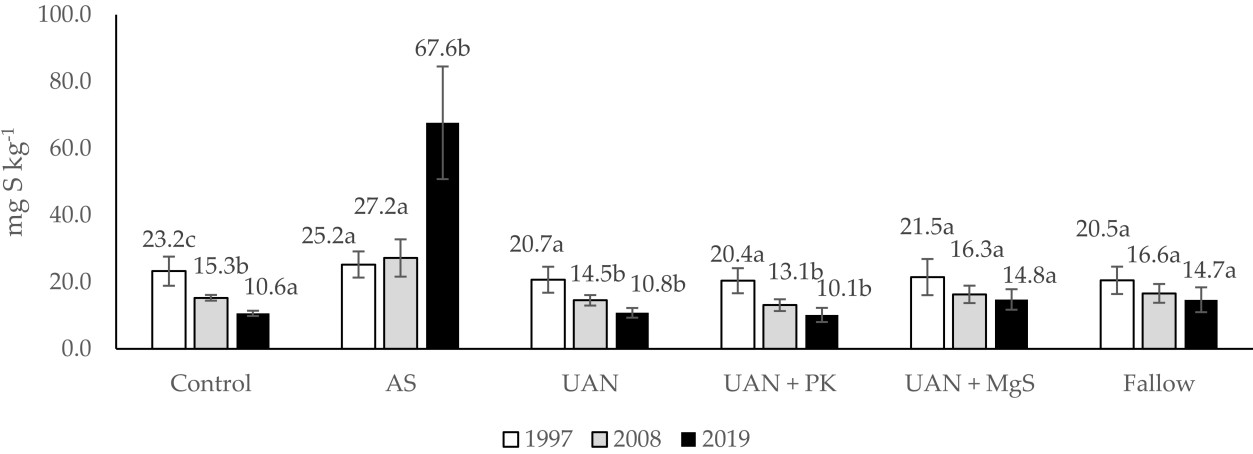

**Figure 4.** Content of Mehlich 3 extractable sulfur (mg S kg$^{-1}$) in topsoil; the lowercase letters describe statistically significant differences between years. Tukey HSD test ($p < 0.05$).

Since S$_{av}$ is calculated as S$_w$ + S$_{ads}$, the contribution of each fraction to S$_{av}$ was calculated. For Cont, UAN, UAN + PK, UAN + PMgS, and Fallow, the contribution of S$_w$ to S$_{av}$ was 76% on average in 1997, and 62% in 2019. The only exception was the AS

treatment in all fractions that accumulated S, where the increase ranged from 79% in 1997 to 82% in 2019.

The $S_{av}$ fraction of the Cont treatment amounted to 16.2, 8.5, and 6.9% of $S_t$ in 1997, 2008, and 2019, respectively. $S_{av}$ for the Fallow treatment made up 16.5, 10.1, and 9.4% in 1997, 2008, and 2019, respectively. UAN, UAN + PK, and UAN + PMgS behaved very similarly to Cont.

Interestingly, it seems that the $S_t$ content on the Fallow treatment seems to have stabilized in 2008 and almost identical values are measured in 2019, while the Cont treatment shows a lowering content.

Extraction with Mehlich 3 (Figure 4) produced results resembling those in Figures 2a,b and 3a, with UAN + PMgS and Fallow treatments showing no significant differences in S content. Pearson correlation was calculated between $S_{av}$ and $S_{M3}$ fractions resulting in a correlation coefficient of 0.958 and 0.990 (both at $p < 0.001$) in 1997 and 2019, respectively. Including all values for $S_{av}$ (Figure 3a) and $S_{M3}$ (Figure 4), it is possible to calculate the average content in each fraction at 26.0 and 20.0 mg S kg$^{-1}$ for $S_{av}$ and $S_{M3}$. These numbers are not significantly different according to the Tukey HSD test at $p < 0.05$. It can be argued that mainly the content of S on the AS treatment in both fractions in 2019 distorts this comparison by increasing the standard deviation. After removing all outlying values from both fractions, we obtain averages of 20.9 and 16.0 mg S kg$^{-1}$ for $S_{av}$ and $S_{M3}$. Even these results produce no significant difference under the Tukey HSD test at $p < 0.05$. This statistical similarity and close correlations among $S_{M3}$ and $S_{av}$ allow us to use $S_{M3}$ values for estimating the S content.

Table 7 presents the relative S contents measured in different soil S fractions comparing 2019 S levels with 1997. It is obvious that (aside from AS) all treatments suffered a decrease of S in time. In 2019, the remaining $S_w$, $S_{ads}$, $S_{av}$, and $S_t$ pool of Cont was 23, 58, 31, and 73%, respectively. UAN, UAN + PK, and UAN + PMgS show a similar decrease of S fractions. On the other hand, AS showed an increase in $S_w$, $S_{ads}$, $S_{av}$, and $S_t$ fractions to 233, 193, 224, and 101% in comparison to 1997.

**Table 7.** Relative S content (%) in individual fractions comparing actual S content in 2019 with 1997 (S content in 1997 = 100%).

| Treatment | $S_w$ | $S_{ads}$ | $S_{av}$ | $S_t$ |
|---|---|---|---|---|
| | Mean ± SD | Mean ± SD | Mean ± SD | Mean ± SD |
| Cont | 23 ± 8.00 | 58 ± 9.40 | 31 ± 9.44 | 73 ± 5.05 |
| AS | 233 ± 62.60 | 193 ± 47.73 | 224 ± 58.54 | 101 ± 13.37 |
| UAN | 33 ± 10.12 | 73 ± 11.40 | 43 ± 10.65 | 77 ± 10.50 |
| UAN + PK | 30 ± 5.06 | 59 ± 10.55 | 37 ± 6.06 | 79 ± 2.94 |
| UAN + PMgS | 42 ± 5.51 | 74 ± 11.84 | 50 ± 7.10 | 76 ± 5.73 |
| Fallow | 45 ± 9.51 | 65 ± 8.12 | 50 ± 9.41 | 88 ± 4.98 |

### 3.3. Subsoil Sulfur Content

Subsoil S content was also evaluated for $S_w$, $S_{ads}$, $S_{av}$, and $S_{M3}$ fractions (Figure 5) and $S_t$ (Figure 6). Comparison of different treatments was performed only for the year 2019. Content of $S_w$, $S_{ads}$, and $S_{av}$ fractions shows the same tendencies. Cont, UAN, and UAN + PK are statistically comparable, while S content measured in UAN + PMgS is significantly higher. Content in AS is significantly higher even than that of UAN + PMgS. The $S_t$ content on the UAN treatment shows the lowest values, significantly lower than Cont, AS, and Fallow. Cont, AS, and UAN + PMgS and Fallow are statistically comparable, but generally the entire Figures 2a,b, 3a, 4 and 5 show that the AS treatment tends to have wider error bars than the other treatments.

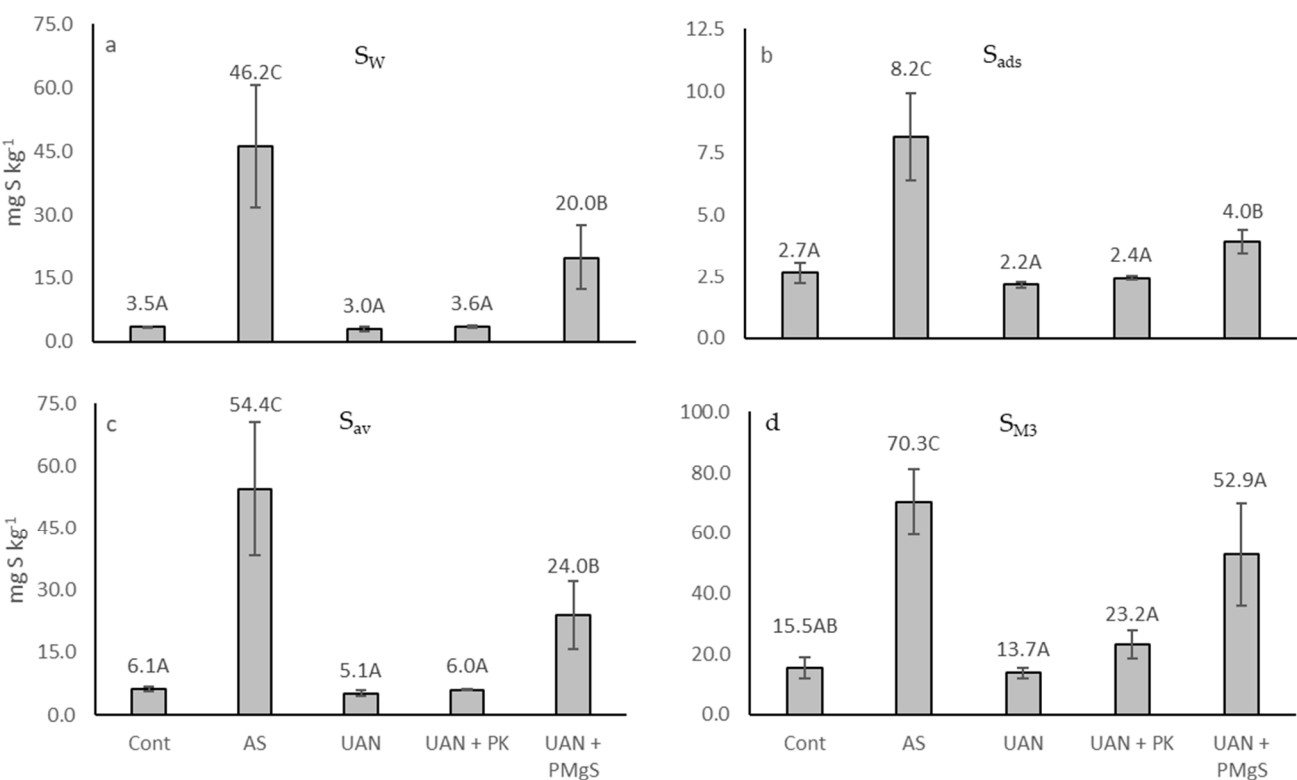

**Figure 5.** Content of sulfur fractions in subsoil (mg S kg$^{-1}$) in 2019: $S_w$ (**a**); $S_{ads}$ (**b**); $S_{av}$ (**c**); $S_{M3}$ (**d**). The capital letters describe statistically significant differences between treatments. Tukey HSD test ($p < 0.05$).

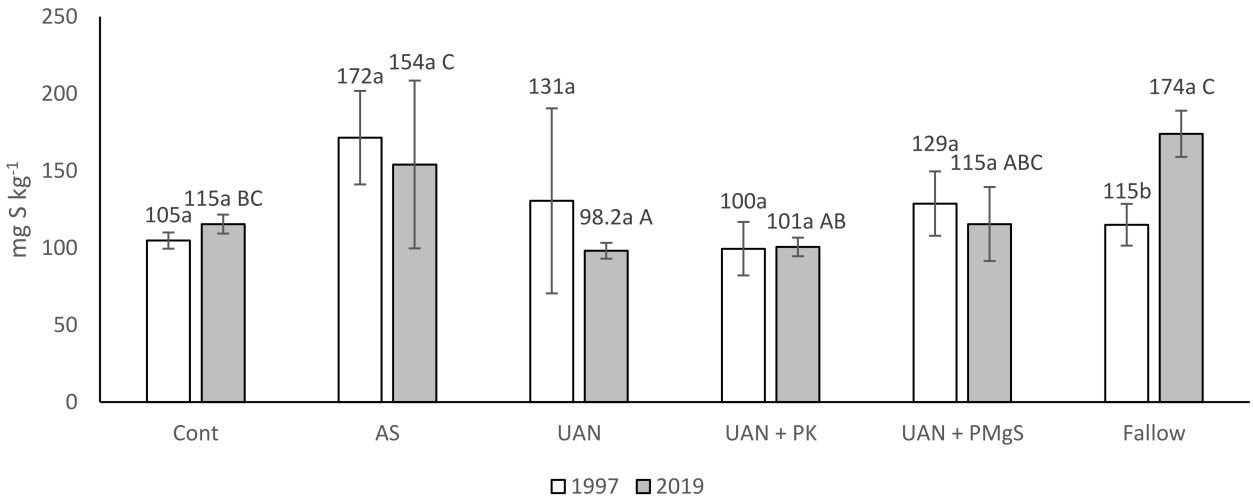

**Figure 6.** Content of pseudo-total sulfur ($S_t$) fraction in subsoil (mg S kg$^{-1}$) in 1997 and 2019; the lowercase letters describe statistically significant differences between years and the capital letters (for 2019) between treatments. Tukey HSD test ($p < 0.05$).

The calculation of different ratios resulted in the following: $S_w$ contributes to $S_{av}$ by 57, 85, 59, 60, and 83% in Cont, AS, UAN, UAN + PK, and UAN + PMgS treatments, respectively. $S_{av}$ makes up 5.3, 35, 5.2, 5.9, and 21% of $S_t$ in Cont, AS, UAN, UAN + PK, and UAN + PMgS treatments, respectively.

AS and UAN + PMgS in $S_{M3}$ treatments show significant increase above other treatments. UAN + PK also seems to be different to UAN. On average, the content of $S_{M3}$ (35.1 mg S kg$^{-1}$) is higher than that of $S_{av}$ (19.4 mg S kg$^{-1}$). There is no significant difference according to ANOVA with Tukey HSD post-hoc test at $p < 0.05$; however, the

standard deviation for $S_{av}$ is so wide (even after removal of AS from the calculation) that this comparison is not appropriate.

Enough data were present to evaluate $S_t$ content not only among treatments, but also its change in years comparing 1997 with 2019 (Figure 6). Cont, UAN + PK, UAN + PMgS, and AS treatments led to no significant change, while Fallow showed an increase in S content. Error bars present on the UAN treatment do not allow for accurate comparison.

### 3.4. Comparison of Sulfur Status in Topsoil and Subsoil

It is clear that in the topsoil in 1997, the results of almost all treatments were similar; however, in 2019, the AS shows significantly higher values over Cont, UAN, and UAN + PK treatments (capital letters), while being comparable with UAN + PMgS and Fallow. Cont, UAN, and UAN + PK are not different between each other, but they also have some overlap with UAN + PMgS and Fallow.

In the subsoil in 1997, S content in the AS treatment was already significantly higher than that of other treatments. Cont, UAN + PK, UAN + PMgS, and Fallow are all comparable. The year 2019 in subsoil is the most interesting, as UAN and UAN + PK produced the lowest values of S content, significantly lower than Cont. Cont itself seems to be similar to AS and UAN + PMgS treatments. Fallow produced the highest value of S content in subsoil, statistically comparable to AS.

Differences in S content at the beginning of the trial and its end were also evaluated (Table 8). In subsoil, the only difference in S content between the beginning and the end of the experiment was detected in the Fallow treatment.

**Table 8.** Balance of S in topsoil and subsoil comparing year 1997 to 2019 (kg S ha$^{-1}$).

| Treatment | Cont | AS | UAN | UAN + PK | UAN + PMgS | Fallow |
|---|---|---|---|---|---|---|
| | Mean ± SD | Mean ± SD | Mean ± SD | Mean ± SD | Mean ± SD | Mean ± SD |
| | | | Topsoil | | | |
| 1997 | 952 ± 86 A b | 900 ± 42 A a | 885 ± 52 A b | 845 ± 97 A b | 943 ± 71 A b | 872 ± 75 A a |
| 2019 | 692 ± 74 A a | 911 ± 105 B a | 682 ± 74 A a | 669 ± 82 A a | 716 ± 69 AB a | 769 ± 67 AB a |
| Balance | −260 | 12 | −203 | −176 | −227 | −103 |
| | | | Subsoil | | | |
| 1997 | 471 ± 41 A a | 772 ± 109 B a | 587 ± 56 AB a | 448 ± 32 A a | 579 ± 63 A a | 519 ± 42 A b |
| 2019 | 519 ± 26 B a | 693 ± 123 BC a | 442 ± 208 A a | 453 ± 79 A a | 520 ± 84 AB a | 785 ± 23 C a |
| Balance | 48 | −78 | −146 | 5 | −60 | 266 |

The capital letters describe statistically significant differences between treatments. The lowercase letters describe statistically significant differences between years. Tukey HSD test ($p < 0.05$).

The balance row shows how much sulfur each treatment gained (+) or lost (−) over the course of the trial. In topsoil, the only influx of S was registered on the AS treatment (although only 12 kg of S ha$^{-1}$), while others lost some of their initial content (ranging between 103 to 260 kg of S ha$^{-1}$). In subsoil, the Fallow treatment gained 266 kg of S ha$^{-1}$, while Cont gained 48 kg S ha$^{-1}$ and UAN + PK gained only 5 kg S ha$^{-1}$. UAN, AS, and UAN + PMgS lost 146, 78, and 60 kg S ha$^{-1}$, respectively.

### 3.5. Sulfur Movement in Topsoil and Subsoil

Table 9 provides an insight into the movement of sulfur in soil. The highest losses of sulfur were present in the AS treatment, followed by the UAN + PMgS treatment losing 3193 kg and 2133 kg of S, respectively, in both topsoil and subsoil. Other treatments lost much less, 204 to 356 kg of S ha$^{-1}$. In subsoil, losses were 199, 308, and 367 kg of S ha$^{-1}$ during the entire experiment for UAN + PK, UAN, and Cont, respectively. Fallow in subsoil lost only 11 kg of S ha$^{-1}$. The period 1997 to 2019 was selected as there is a more representative dataset allowing us to perform this calculation.

**Table 9.** S movement in topsoil and subsoil during the experiment.

| Treatment | Cont | AS | UAN | UAN + PK | UAN + PMgS | Fallow |
|---|---|---|---|---|---|---|
| | Topsoil (kg S ha$^{-1}$) | | | | | |
| Status in 1997 + inputs 1997–2019 (a) | 1127 | 4340 | 1060 | 1020 | 3050 | 1047 |
| Status in 2019 + uptake 1997–2019 (b) | 771 | 1147 | 838 | 816 | 917 | 769 |
| Loss from topsoil (a − b) | 356 | 3193 | 221 | 204 | 2133 | 277 |
| | Subsoil (kg S ha$^{-1}$) | | | | | |
| Status in 1997 + loss from topsoil (c) | 827 | 3965 | 808 | 651 | 2713 | 796 |
| Status in 2019 (d) | 519 | 693 | 442 | 453 | 520 | 785 |
| Loss from subsoil (c − d) | 308 | 3271 | 367 | 199 | 2193 | 11 |

## 4. Discussion

### 4.1. Biomass Yield and S Uptake

Figure 1a shows relative yields increased in all fertilized treatments. Except for Cont, all fertilized treatments received uniformly 120 kg N ha$^{-1}$ year$^{-1}$ and produced comparable yields, while only two treatments received S fertilizer. Based on the fact, that all N fertilized treatments were statistically the same, we can deduce that S was not a limiting nutrient in our trials.

Relative sulfur uptake (Figure 1b) showed more differences, as treatments amended with S fertilizer have the highest relative S uptake. Cont treatment provided uptake of 3.38 kg S ha$^{-1}$ year$^{-1}$. Meanwhile application of 142 kg S ha$^{-1}$ year$^{-1}$ in the form of AS resulted in uptake of 9.84 kg S ha$^{-1}$ year$^{-1}$, while application of UAN + MgS (84 kg S ha$^{-1}$ year$^{-1}$) increased S uptake to 8.31 kg S ha$^{-1}$ year$^{-1}$ (increase by 291% and 246% over Cont, respectively). A similar increase of S uptake was present in the study of Sakal et al. [38], where a dose of 40 kg S ha$^{-1}$ increased S uptake of maize by up to 9.44 kg S ha$^{-1}$ over unfertilized treatment, which provided uptake of 2.58 kg S ha$^{-1}$ in trials with maize and wheat crop rotation. Overall, the increase in S uptake was also noted on treatments UAN and UAN + PK, providing increase of 192% and 180%, respectively. Weil and Mughogho [30] presented similar results, as they also observed increase in S uptake on treatments fertilized by 80 kg of N ha$^{-1}$. Generally, the application of N increases yields and, thus, produces higher uptake of other nutrients [39], including S [40].

Table 4 describes these results in more detail. Interestingly, uptake on each treatment shows descending tendencies in periods 2002–2007, 2008–2013, and 2014–2019. During these periods, yields on individual treatments also show a significant decrease (not present in this study); therefore, reduced uptake can be attributed to reduced yields. The reduction of yields over time can be partially attributed to N dose. Our trials were previously investigated by Černý et al. [41], where they focused on N uptake. They found that uptake of N can reach up to 185 kg N ha$^{-1}$, while yearly inputs are 120 kg N ha$^{-1}$, so the nitrogen dose might not be high enough to properly supply the crops. Other possible explanations can include a change of the crop hybrid (viz. materials and methods) and precipitation, which can cause mobilization of N in soil and possibly N leaching [42,43], as well as optimal growth of the crop itself and better nutrient availability [44].

Given the results of correlation analysis (Table 5), it seems that the S$_{ads}$ fraction correlates best with total S content in plant biomass (r = 0.747; $p < 0.01$), followed closely by S$_{av}$ (r = 0.725; $p < 0.01$) and S$_W$ (r = 0.674; $p < 0.01$). S$_{av}$ is calculated as the sum of S$_W$ and S$_{ads}$, thereby explaining the very similar coefficients of these fractions. This is generally in agreement with other works, stating that S$_W$, S$_{ads}$, and their sum Sav are very important fractions in plant nutrition [17,35] (as discussed later). S$_{M3}$ correlates only moderately with the content of S in biomass. Since Mehlich 3 includes HNO$_3$, it is possible that it extracts some part of the S$_t$ content that is usually not available to plants. Relationship between the mentioned fractions and S biomass content may not represent the reality well enough in this study due to the small sample set size. Recently, Sedlář et al. [45] correlated content of S in

maize biomass with content of S in $S_W$ and $S_{M3}$ fractions at Pearson´s correlation coefficient values of r = 0.961 and 0.804, respectively, at $p < 0.001$ in a pot experiment. In field trials, correlation coefficients were measured at r = 0.174 (insignificant) and 0.629 ($p < 0.05$) for $S_W$ and $S_{M3}$, respectively. In their experiments, S biomass content was determined early during vegetation (BBCH 16–18), while in this study, maize was sampled quite a bit later (BBCH 75). Differences in sampling period can explain some of the differences, as content in S is quite variable in soil and plants across the vegetation season [46]. In addition, Sedlář et al. [45] collected maize from farmers scattered across the Czech Republic using both a crop rotation system and monoculture, bringing even more variability into the evaluation. On the other hand, this study focuses solely on maize from long-term monoculture on one station only.

### 4.2. Topsoil and Subsoil Sulfur Content

Results of two-way ANOVA (Table 6) revealed significant interaction of year and treatment on content of S in all soil S fractions, with the exception of $S_t$. This was caused by a high content of S on AS treatment in the soil in the year 2019 (Figures 2 and 3), which resulted in significant interaction of (AS treatment × year 2019 pair). Possible causes of such increase in content of S on this treatment are discussed later. These results would suggest that once inputs of S reach such levels, as with the inputs of S on AS treatment in our study (Table 3), the S starts to cumulate in $S_w$, $S_{ads}$, $S_{av}$, and $S_{M3}$ fractions over time. UAN + PMgS treatment received S fertilizer as well. Interactions of (UAN + PMgS × any year pairs) were rarely significant and, in almost all cases, not different from pairs with (treatments without S fertilizer and any year). This suggests that the fertilizer rate on UAN + PMgS is not high enough to cause cumulation of S in topsoil.

Soil samples were collected after crop harvest. During crop growth, levels of S (especially $S_{av}$) would be high enough for sufficient crop growth in spring. Grobler et al. [47] mention that maize is capable of producing sufficient yields with 10 mg $S_{av}$ kg$^{-1}$ of soil. $S_{av}$ in our trials (Figure 3a) never reached lower values than 10 mg $S_{av}$ kg$^{-1}$ even after harvest (on average around 44 kg $S_{av}$ ha$^{-1}$), when S pools were already drained. Considering the results of $S_{M3}$ extraction (Figure 4 in topsoil and Figure 5d in subsoil), $S_{M3}$ extracted a very similar amount of sulfur to $S_{av}$. Kulhánek et al. [13] investigated the relationship between $S_{av}$ and $S_{M3}$ and found a very close Pearson correlation (r = 0.882; $p < 0.001$). Using the Pearson correlation analyses, we also found a very close relationship of these two fractions in this study (r = 0.958 in 1997 and 0.990 in 2019, both at $p < 0.001$). This high correlation can be caused by the fact that each treatment received the same amount of fertilizer each year for the entire duration of the trials, as well as all treatments of this study being located on the same soil type, while Kulhánek et al. [13] investigated a wider variety of soil types with much more varied fertilizer inputs. A similar relationship was also reported in other studies [48,49]. As mentioned in the Results, no differences between $S_{av}$ and $S_{M3}$ in topsoil and subsoil (Tukey; $p < 0.05$) were found. This makes the Mehlich 3 method suitable for evaluating available S content in soil.

Kulhánek et al. [50]—in Czech—developed a certified methodology for determination of plant available S using Mehlich 3 extractant. They proposed criteria (5 categories in total) for evaluation of plant available S using $S_{M3}$ content in soil. The results of our study place $S_{M3}$ content in topsoil in the "Satisfactory" category in 1997. Having said this, UAN and UAN + PK treatments have 20.7 and 20.4 mg S kg$^{-1}$, which technically puts them on the border of the "Low" and "Satisfactory" categories (ranges for "Low" and "Satisfactory" categories are 11–20 and 21–30 mg S kg$^{-1}$). On the other hand, in the year 2019 the $S_{M3}$ content is ranked in the "Low" category in all treatments, since the content of $S_{M3}$ has decreased in general with the exception of the AS treatment. In subsoil, there is no temporal comparison, since we only have results from 2019 (Figure 5d). Using the categories proposed by Kulhánek et al. [50], Cont and UAN are in the "Low" category, UAN + PK is ranked in the "Satisfactory" category, and UAN + PMgS and AS are ranked in the "High" content category.

Overall, S pools of different S fractions on all treatments in topsoil seem to be decreasing in time with the exception of the AS treatment (Figures 2 and 3), which shows an increase in $S_w$, $S_{ads}$, and $S_{av}$ fractions, while maintaining even levels of $S_t$.

It is obvious that (aside from AS) all treatments suffered a decrease of S pools over time (Table 7). In 2019, the $S_w$ remaining pool in the Cont, UAN, and UAN + PK treatments ranged from 23 to 33% of $S_w$ determined in the samples of the year 1997. $S_{ads}$ content on these treatments ranged 58–73% as compared to 1997 and in 2019. $S_{av}$ content ranged from 31–43% of $S_{av}$ in 1997. Lastly, $S_t$ content for Cont, UAN, and UAN + PK treatments in our trials in 2019 reached 73–79% of those in 1997. This conforms to Balík et al. [9], who investigated sulfur status in soils on 10 different sites in the Czech Republic. They found that $S_w$, $S_{ads}$, $S_{av}$, and $S_t$ content decreased in time to 32, 61, 50, and 92%, respectively (comparing the year 1981 with 2007). Changes in mineral S fractions were evaluated by Kulhánek et al. [19] in a long-term field experiment (comparing the year 1996 with 2014) with a potato, wheat, and barley crop rotation and several fertilizers. They describe a decreasing tendency of $S_{av}$ content in the entire trial; however, $S_w$ content increased following the application of sewage sludge after potatoes. In other crops, $S_w$ always showed a decrease.

Indeed, the application of organic matter into soil can increase its water holding capacity [51] and, therefore, temporarily increase the $S_w$ fraction. In our experiment, no organic fertilizers were applied to the soil during the entire experiment. Changes of $C_{SOM}$ in our monoculture trial were investigated by Balík et al. [52]. Initial $C_{SOM}$ levels decreased from 1.26% in 1993 to 0.98% in Cont and 0.93% in AS in topsoil in 2019. Subsoil $C_{SOM}$ was reported at 0.73% in Cont and 0.65% in AS in 2019. This is caused by cultivation of maize for silage, since very little post-harvest residues are present (just stubble and roots), and $C_{SOM}$ presents net loss. Considering the current content of $C_{SOM}$ in Cont topsoil (0.98%) and subsoil (0.73%) [52], it is possible to calculate their ratio at 1.34 as well as a ratio of current $S_t$ levels in topsoil (157 mg S kg$^{-1}$–Figure 3b) and subsoil (115 mg S kg$^{-1}$–Figure 6), where we get the ratio 1.36. It seems that $S_t$ content is bound in the same way in organic matter in topsoil and subsoil. After all, organic bound S can make up to 90% of $S_t$ [12]. Loss of the $S_{av}$ fraction from 1997 to 2019 in topsoil (Figure 3a) amounted roughly to 16.2–24.0 mg S kg$^{-1}$). Loss of $S_t$ (Figure 3b) ranged from 24 to 59 mg S kg$^{-1}$ in all treatments except AS. This suggests that loss of S from the organic fraction was present and partially contributed to loss in $S_t$. In arable systems where soil organic matter is not accumulating, there is little opportunity for inorganic $SO_4^{2-}$ ions immobilization into organic matter [53]. The $S_t$ loss can probably be attributed to leaching of the $SO_4^{2-}$ anion, which is discussed later.

Even the Fallow treatment shows interesting results. Content in all S fractions is highest in 1997, but lower in 2008. This treatment has no crop production, so the decrease in the period 1997–2008 can be attributed to leaching of S from atmospheric depositions. Unlike the other non-AS treatments, the decrease of S was not present from 2008 to 2019. The most likely cause is the decrease of atmospheric S depositions in recent years [10]. These results are similar to those of Gourav et al. [33], where a higher content of $S_w$ and $S_t$ was present on the Fallow treatment than on Cont after 48 years of experiment with maize monoculture. In our study, the Cont treatment of the $S_{av}$ fraction amounted to 16.2, 8.5, and 6.9% of $S_t$ in 1997, 2008, and 2019, respectively. $S_{av}$ for the Fallow treatment made up 16.5, 10.1, and 9.4% in 1997, 2008, and 2019, respectively. UAN, UAN + PK, and UAN + PMgS behaved very similarly to Cont. It is clear, that treatments with maize production and S leaching had a higher influence over the decrease of $S_{av}$ pools than that of Fallow, which is only influenced by leaching.

The AS treatment seems to show opposing results. The increase of $S_{ads}$ in topsoil in this treatment (Figure 2b) can be explained, e.g., by a decrease of pH that causes increased sorption of $SO_4^{2-}$ [54]. AS fertilization indeed decreases the pH of soil [55]. This, however, does not explain the increase of $S_w$ and $S_t$ fractions. A possible explanation may be the combination of site and rainfall variability as well as annual application of 142 kg S per hectare (which can be considered as a very high dose), which could cause the accumulation

of sulfur in this treatment over time. Influence of time was deemed significant earlier (Table 6) and seems to support this thought.

From the S content in all fractions (Figure 5) in subsoil was increased in 2019 on UAN + PMgS and AS treatments over those without S fertilization. Knights et al. [53] and Zayed et al. [56] report that surplus S can be accumulated in subsoil, which could explain that the highest increase overall was present on the AS treatment that received the highest amounts of S inputs, followed by UAN + PMgS, which received the second highest inputs of S. This can also explain the increased content of $S_t$ in 1997 on the AS treatment (Figure 6). Even though we evaluate data from 1997, the experiment and fertilizer applications started in 1993, so AS was applied for several years before the first data are presented in this study. Leaching of S from previous years could have increased S content in the subsoil in 1997 on this treatment in subsoil.

$S_w$ content in subsoil on AS and UAN + PMgS treatments had increased by 42.7 and 16.5 mg S kg$^{-1}$ over Cont (Figure 5a). Since soil samples were taken after harvest, it is possible that soil was sampled during a downward movement of S, which was in surplus in topsoil. This could also explain the increase in $S_{ads}$ (Figure 5b). The $S_{av}$ (Figure 5c) content on Cont, UAN, and UAN + PK is on par; however, $S_t$ content (Figure 6) is somewhat lower for UAN and UAN + PK in comparison with Cont. It is possible that a higher uptake of S by plants in the UAN and UAN + PK treatments could cause mineralization and upward movement of S from the subsoil [51]. Accumulation of total S over time is visible in the subsoil on Fallow in comparison with other non-S-fertilized treatments (Figure 6). Since there is no crop production on Fallow, we assume that maize roots have influenced the other treatments. Indeed, maize roots can grow as deep as the subsoil layer (30–60 cm) [57,58] and could, therefore, cause a decrease in subsoil S content.

*4.3. Sulfur Status and Movement in Topsoil and Subsoil*

Values of $S_t$ in the topsoil and subsoil (Table 8) are calculated in kg S ha$^{-1}$. Therefore, temporal changes represent identical tendencies for the soil S content in Figures 3b and 6.

From Table 8, it is clear that all treatments except AS show a decrease in $S_t$ content in the topsoil. Negative S balance can be attributed to crop growth and leaching [24]. Cont is influenced by leaching and maize growth, while Fallow is only influenced by leaching; therefore, it is understandable that Fallow showed the lowest loss of S. A slightly positive balance is noticeable in AS. A possible explanation may be the combination of site and rainfall variability as well as annual application, as was mentioned above in the discussion of topsoil and subsoil sulfur content. In the subsoil Cont, Fallow, and UAN + PK treatments, we measured the increase in S content. It is possible that leached sulfates from the topsoil were absorbed or accumulated in organic matter.

Table 9 describes the movement of S in the topsoil and subsoil. The calculation presented in this table is based on a comparison of the original S status in soil and total S inputs during trials with the status of S at the end of the experiment with total plant uptake during the experiment. Using this method, we calculated that on all treatments in topsoil, an extensive S loss is present. In subsoil, the trend is the same, with the exception of the Fallow treatment that seems to have stable values. We consider the loss from subsoil in Table 9 equal to the total S lost from the topsoil and subsoil profile (0–60 cm depth). The Fallow treatment registered the smallest loss of 11 kg S ha$^{-1}$, while Cont, UAN, and UAN + PK lost 308, 367, and 199 kg S ha$^{-1}$ during the entire trial, respectively. UAN + MgS and AS provided losses of 2193 and 3271 kg S ha$^{-1}$ during the entire trial or 95.3 and 142 kg S ha$^{-1}$ year$^{-1}$, respectively. This loss can certainly be attributed to leaching of $SO_4^{2-}$ anions as described in Riley et al. [24], where a three-year pot experiment with AS application and undisturbed soil sample was established with a focus on S leaching. Here, the control treatment received a total input of 22 kg S ha$^{-1}$ in terms of depositions, while the AS treatment received an additional 50 kg S ha$^{-1}$ in the form of AS fertilizer (in total 72 kg S ha$^{-1}$ was added). It was found that 78% of S applied on AS was leached in the first year and it increased to 96% after three years. When loss by leaching and plant

uptake were added together (55 and 107 kg S ha$^{-1}$ in Cont and AS, respectively), the total S outputs in AS and even Cont treatments exceeded the total inputs at the end of the trial by 35 and 33 kg S ha$^{-1}$, respectively. The authors explain this by net S mineralization from organic bound S in soil. Using these data, we calculated a ratio (in%) of outputs (uptake and leaching) over inputs (deposition and fertilizer) and obtained 253% for Cont and 149% for AS. The same calculation was performed for Cont, AS, and UAN + PMgS treatments with the result of 221%, 102%, and 114%, respectively. Furthermore, S uptake (Table 4) did not reach the levels of fertilizer inputs (Table 3) and mobile sulfates were generally susceptible to percolation in AS and UAN + PMgS treatments. Leaching of sulfates was obviously present. Riley et al. [24] measured the presence of organic bound S in leachates. It makes sense in the context of our study, since Balík et al. [52] evaluated changes in organic bound S on the same trials as this study and found a decrease of organic matter content that reduces space for net $SO_4^{2-}$ accumulation. This lack of S retention was also demonstrated in Rothamsted long-term experiments [51], where applications of 52–220 kg S ha$^{-1}$ year$^{-1}$ over 150 years had not increased organic, nor inorganic, pools of S. In addition, we demonstrated (Figure 3b) that $S_t$ content in our trials is decreasing and organic bound S is, of course, part of this fraction [17].

Bergholm and Majdi [25] also report S leaching during a six-year experiment. However, their results differ from those of ours. Out of a total S input of 821 kg S ha$^{-1}$ in the form of depositions and AS fertilizer, up to 339 kg S ha$^{-1}$ was leached (41.2%) in comparison with Cont, where inputs were 133 kg S ha$^{-1}$ in the form of atmospheric S depositions, and leaching amounted to 79 kg S ha$^{-1}$ (59%). Accumulation in AS and Cont reached 317.1 and 8.1 kg S ha$^{-1}$, respectively. Calculation of outputs (leaching and accumulation) over inputs ratio was performed resulting in 60% and 65% for AS and Cont treatments, respectively. These results differ from those in our study or in Riley et al. [24]. This difference is due to the fact, that the Bergholm and Majdi [25] study was conducted on forest soil. Authors measured accumulation of S in the forest floor (that is very rich in organic matter) and also in mineral soil. On the other hand, Riley et al. [24] as well as our study conducted an experiment on arable soil. They found leaching of organic bound S and a recent study by Balík et al. [52] found a decrease of soil organic matter on our trials. There are other factors that can influence the movement of S in soil, like an increase in precipitation being capable of increasing movement of nutrients in soil [26,27]. Soil texture can have an influence on water retention and, therefore, leaching as well. The soil texture in our trials was Loam with 5.4% clay content. In Riley et al. [24], the soil texture was Sandy loam of 0–40 cm depth with clay content around 7.5% and Loamy sand of 40–60 cm with clay content around 5.5%. In Bergholm and Majdi [25], the soil texture was also Loamy sand with 4% clay. In the end, it is possible to state that the majority of the disparity can be attributed to vastly different conditions between forest and arable soil.

Comparing the results of our maize monoculture long-term field experiment with a pot experiment [24] and forest trial [25] is indeed difficult due to the very different conditions that apply to each of these respective environments. Even though these trials are so different, what they share in common is the fact that a significant part of sulfur fertilizers is leached and unavailable to plants and may even be contaminating ground waters [59].

In general, to reduce the leaching of sulfates, it is recommended to use fertilizers with lower solubility, like elemental S, which is less susceptible to leaching [24,60], or performing the application in several smaller doses [24]. However, Santoso et al. [60] mention that fast growing crops like maize show significantly lower S uptake if fertilized by slowly soluble fertilizers, especially if applied in split doses.

## 5. Conclusions

In this study, we examined the influence of S fertilization and atmospheric depositions on soil and plants in maize monoculture lasting 27 years. We found that S was not a limiting nutrient and S fertilization did not increase yields, although it created higher S uptake by maize. Generally, we noticed a decrease in topsoil $S_w$, $S_{ads}$, $S_{av}$, and $S_t$ content on

all treatments with exception being the AS treatment, which doubled its initial content in mineral fractions and maintained the same levels of $S_t$, which we attribute to the very high S dose on this treatment. In our conditions as well as similar ones, $S_{av}$ fractions ($S_w + S_{ads}$) determination seems to be most suitable soil tests for plant S uptake prediction.

Using the total S balance, we demonstrate that a significant amount of S is being leached below the soil depth of 60 cm on all treatments over the period of 27 years of the experiment, with the highest values on both S fertilized treatments. We conclude that high atmospheric depositions before the 1990s caused S pools in soil to fill up and contribute to S leaching. Furthermore, the decrease of S depositions in the period of the 1990s to 2019 and leaching of S caused a decrease in S content in the topsoil in almost all treatments (except AS) during the period of 1997–2019. Based on our results, we can propose the use of multiple small S doses or slowly soluble fertilizers as well as organic fertilizers in maize monoculture.

**Author Contributions:** Conceptualization, P.S. and M.K.; Formal analysis, O.S.; Methodology, M.K. and J.Č.; Supervision, J.B.; Writing—original draft, P.S. All authors have read and agreed to the published version of the manuscript.

**Funding:** This work was funded by the European Regional Development Fund (ERDF)–Center for the investigation of synthesis and transformation of nutritional substances in the food chain in interaction with potentially harmful substances of anthropogenic origin: Comprehensive assessment of soil contamination risks for the quality of agricultural products (NutRisk Center, Registration number: CZ.02.1.01/0.0/0.0/16_019/0000845).

**Data Availability Statement:** Data available at corresponding author.

**Acknowledgments:** Thanks to the team of the Department of Agroenvironmental Chemistry and Plant Nutrition of Czech University of Life Sciences in Prague for help and support, Czech Hydrometeorological Institute for necessary data and Hana Zámečníková and Jana Najmanová for help with laboratory work and measurements.

**Conflicts of Interest:** Authors declare there is no conflict of interest.

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
