# Peer review of "Evaluation of Soil S Pools under 23 Years of Maize Monoculture"

_agronomy, doi:10.3390/agronomy11122376_

Round 1
Reviewer 1 Report
To the Authors, it is my second time reading your work. This time you improved your nicely written paper. I found the answer to all questions from the firths round were satisfying. The new information and the clarifications on the text are well done. The problem was well explained and connected over the introduction. Treatments, replicates, analyses, descriptions, and statistical designs are nicely presented and improved. The assumptions of the S deposit are well justified. The results section was clear and followed the figures. Meanwhile, the discussion gives relevant statements supported by according literature. The conclusion is clear and directly related to the paper's aim.
I will like to acknowledge the authors of this article. because this paper has really clear ideas, well-described, nicely developed results with a complete discussion with a clear conclusion of the work.
Some minor mistakes I found was
In the abstract, the format of the unit is not on the super index
The form of letters to individual statistical differences are different between tables. Table 8 has letters with superindex format. Meanwhile, other tables are standard size.
In conclusion, line 651 is the phrase. "In our or similar conditions," the words "our or" I do not know if is correct, it could be "in our conditions as well as similar ones". I understand the phrase but had to read it two or three times.
Author Response
Thank you.
Corrected mistake in abstract regarding indexing of units.
Corrected and unified indexing of tables throughout entire document.
Corrected sentence on line 651 using line suggested by you.
Reviewer 2 Report
The SD in the table should be shown as mean±SD. while in present table, the SD was shown in a separate column, which was difficult to read.
Author Response
Corrected tables to make them more readable as suggested by you.